# Data-driven learning of structure augments quantitative prediction of biological responses

Yuanchi Ha[1,2], Helena R. Ma [1,2], Feilun Wu[1], Andrea Weiss[1], Katherine Duncker[1,2], Helen Z. Xu[1], Jia Lu[1,2], Max Golovsky[1], Daniel Reker[1,2], Lingchong You[1,2,3] *

1 Department of Biomedical Engineering, Duke University, Durham, North Carolina, United States of America, 2 Center for Quantitative Biodesign, Duke University, Durham, North Carolina, United States of America, 3 Department of Molecular Genetics and Microbiology, Duke University School of Medicine, Durham, North Carolina, United States of America

* you@duke.edu

## Abstract

Multi-factor screenings are commonly used in diverse applications in medicine and bioengineering, including optimizing combination drug treatments and microbiome engineering. Despite the advances in high-throughput technologies, large-scale experiments typically remain prohibitively expensive. Here we introduce a machine learning platform, structure-augmented regression (SAR), that exploits the intrinsic structure of each biological system to learn a high-accuracy model with minimal data requirement. Under different environmental perturbations, each biological system exhibits a unique, structured phenotypic response. This structure can be learned based on limited data and once learned, can constrain subsequent quantitative predictions. We demonstrate that SAR requires significantly fewer data comparing to other existing machine-learning methods to achieve a high prediction accuracy, first on simulated data, then on experimental data of various systems and input dimensions. We then show how a learned structure can guide effective design of new experiments. Our approach has implications for predictive control of biological systems and an integration of machine learning prediction and experimental design.

## Author summary

Using limited data to predict how biological systems respond to combinations of perturbations is important for better understanding of biological systems and for practical applications. Here we present a new method, structure-augmented regression, that efficiently learns biological response landscapes from sparse measurements. Our method exploits the property that many biological response landscapes have distinct lower-dimensional structures, which can be learned first to assist the subsequent quantitative predictions. We demonstrate that our algorithm outperforms existing algorithms on simulated and experimental data of various biological systems and dimensions. We further exploit the learned structure by suggesting new experiments that refines the learned structure to further improve the prediction accuracy. By integrating machine learning

**Data Availability Statement:** All data and code used for running machine learning and plotting is available on a GitHub repository at https://github.com/YChH/StructureLearning.

**Funding:** This work was partially supported by the National Institutes of Health (L.Y., R01AI125604, R01GM098642, and R01EB031869). The funders had no role in study design, data collection and analysis, decision to publish, or preparation of the manuscript.

**Competing interests:** The authors have declared that no competing interests exist.

with experimental design, our approach has implications for the predictive control of biological systems.

## Introduction

Biological systems respond to a wide range of environmental cues, including temperature [1–3], nutrient variations [4–6], and stresses [7–9]. On one hand, these responses reflect the intrinsic properties of each system; on the other hand, they serve as a tunable property that is being widely exploited [10–12]. This type of manipulation has allowed us to maximize bacterial growth for synthesis of desirable chemicals, such as bioplastic [13–15], biofuels [16–18] and artificial flavors in beer [19,20]. Conversely, one may need to optimize drug combinations (e.g. antibiotics) to eliminate target cell populations [21–23]. In addition, the need for more precise control of sensing, computation and long-term stable production are rising [24,25]. For instance, overexpression of synthetic gene circuits can have deleterious effects on cell viability by imposing too much burden on the system, since these circuits take up limited resources [26,27] or even secret toxic within the host [28,29]. This in turn makes the long-term stable production challenging. In such situations, the ability to fine tune system behavior can enable optimization and greater predictability of the specific system outputs [30–32].

Establishing this predictability is challenging given usually limited experimental data [33,34] or, in clinical operations, limited patient samples [35]. For example, while combination treatment is a promising approach in treating bacterial infections and other complex diseases like cancers [36–38], the number of possible combinations increases exponentially with the number of drugs. For two drugs, exhausting 10 concentrations for each requires testing 100 drug-concentration combinations, which remains feasible. For newer combination therapies that utilize up to 8 drugs [39], exhaustively testing 10 concentrations per drug will require $10^8$ tests, which becomes labor-, material- and cost-prohibitive [40,41].

To improve predictive power using limited data, one approach is to use phenomenological models to guide data analysis [42–44]. A recent study formulated a dose-response model that considers drug interactions enables prediction of 3-drug combination effects using responses of single drugs and of few drug pairs [45]. The key advantage of formulating such a model is to narrow the parameter space by imposing a structure of the dose-response, as reflected by the formulation of model equations, on the data based on the prior knowledge. If the model structure is properly formulated, this approach can be highly data efficient. However, the formulation of these model equations requires sufficient prior knowledge on the system of interest. For example, in this study mentioned above, the key assumption of the model is that the effects of each drug on other drugs in the same combination are multiplicative, and that one can neglect third- and higher-order interactions between drugs. Therefore, such models always raise the question of generalizability.

An alternative approach is machine-learning models that are entirely data driven [33,46,47]. On this front, existing studies mainly combine different available machine-learning (ML) models to generate ensemble predictions by simultaneously taking advantage of all of them, after learning on a small dataset, usually less than 100 datapoints. In one example, multiple general-use regressors are combined, each with a weight, to predict productions of synthetic molecules, including renewable biofuels and hoppy-flavored metabolites, from bioengineered cells, as well as to guide new engineering design to further increase the yields [48]. Designed as a recommendation system, the primary goal of the method is to guide experimental design instead of increasing prediction accuracy. Indeed, while the platform has

successfully recommended new yeast strain design that improved the tryptophan productivity by more than 100%, the prediction accuracy of the outcome alone is wanting. In another example, a regularized boosted regression tree, when combining with composite non-negative matrix factorization, can predict drug combination effect using less training data [47]. This study shows that using this ensemble algorithm, to predict two-drug combination landscape, one only needs single-drug dose responses of both drugs and the diagonal drug combination responses. However, the same study only demonstrates such capability on various two-drug combination applications.

While these ML applications have demonstrated that combining ML methods can improve the prediction accuracy, their results show that reaching a high prediction accuracy remains to be highly data demanding, especially when dealing with combinations of three or more inputs. Drawing inspiration from the use of mechanistic models that impose a structure of the dose-response from prior knowledge that's potentially nonlinear and complex, we wondered if we could exploit the advantage of imposing a structure, without explicitly relying on prior knowledge. Doing so would require us to learn a structure directly from the data.

In response to environmental perturbations, many biological systems exhibit characteristic response landscapes in their outputs, including growth, gene expression, or metabolic functions. The landscapes can often be approximated by a lower-dimensional structural feature. For example, drug pairs can generate different types of dose-response curves on the two-dimensional dose-response landscape. A synergistic pair that improves each other's effect creates concave-down curves; an antagonistic pair that impedes each other's action creates concave-up curves [49]. The whole landscape is filled by similar shaped dose-response curves, each is a certain value away from the previous one. Similarly, a cell population can exhibit characteristic landscapes in response to different combinations of nutrients [13].

We show that the low-dimensional representation of each landscape can be learned from a limited amount of data. Incorporating this learned structure as a soft constraint improves the quality of subsequent data prediction. This new prediction scheme generalizes well to systems that are being tuned by higher-dimensional inputs. Finally, such a learned response structure can be incorporated into an active-learning framework, which consists of iterations of structure learning, quantitative prediction, targeted data collection, and improved structure learning and quantitative prediction. This framework enables data-efficient and accurate construction of the response landscape of a biological system of interest.

## Results

### Learnt structure contains rich information to assist regression

To demonstrate the value of learning structure in regression, we consider a community of two populations, a plasmid-carrying population ($S^1$) and a plasmid-free population ($S^0$) (Fig 1A). $S^0$ cells can turn into $S^1$ cells by receiving a plasmid from $S^1$, and $S^1$ can turn into $S^0$ by losing the plasmid. In the same environment, the two populations compete for a common nutrient, glucose. The rate of gene transfer can be suppressed by a conjugation inhibitor chemical, linoleic acid (Lin). The simulated dynamics reveals a gradual transition of the output, final density of $S^1$, in response to different glucose and Lin concentrations (Fig 1B).

In this simulation, we generated high-density data that reveal a smooth response landscape, which serves as the ground truth for learning (Fig 1B). A common task in experimental analysis is to reconstruct this full landscape from sparsely sampled data. A typical strategy is to do direct regression on the sampled data, which can be sensitive to the sampling variability and learn poorly. Despite the sparsity of sampled data, however, we noted that the key features of the landscape structure can be maintained.

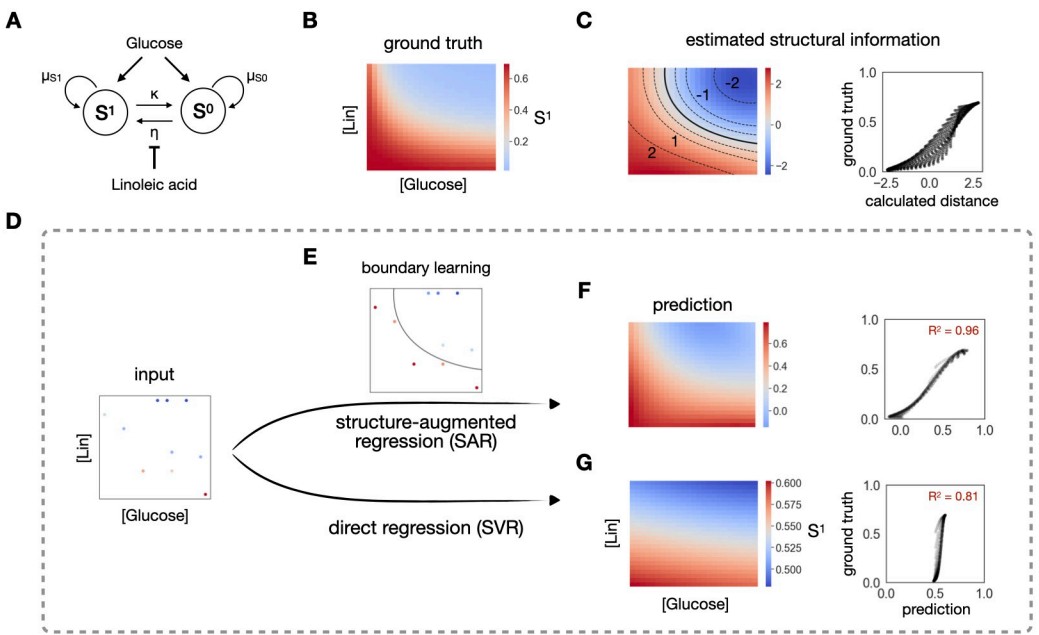

**Fig 1. Structure contains rich information for regression.** A. A simple community of a plasmid-carrying population ($S^1$) and a plasmid-free population ($S^0$). $S^0$ acquires the plasmid through conjugation at rate η, becoming $S^1$. $S^1$ reverts to $S^0$ through plasmid loss at rate κ. The conjugation efficiency η is modulated by an inhibitor, linoleic acid (Lin). The growth rates of both populations are modulated by a common nutrient, glucose. B. Heatmap of the final density of $S^1$ under different concentrations of linoleic acid and glucose. It shows a structured monotonically decreasing response from bottom left to top right on the full simulated landscape. C. Demonstration of the rich structural information. The left-hand side is a heatmap of estimated structural information of $S^1$, calculated as the distance between each point and the boundary, across the landscape. The boundary is denoted by the solid black line. Multiple contour lines of the same distance, denoted as dash lines, are highlighted on top of the heatmap. From bottom to top, these calculated distances are -2, -1, -0.5, 0.5, 1 and 2. The contours of -2, -1, 1 and 2 are labeled on the heatmap. The right-hand side is the scatterplot of the calculated distance and the ground truth over the whole landscape, which serves as a more direct comparison between the estimated structural information and the ground truth. D. Comparison of the two regression methods. The left-most panel shows a training set of 10 data points, sampled from the high-resolution growth truth (A). The top flow shows the scheme of the regression constrained by a learned structure (SAR). This strategy first learns the boundary between high and low $S^1$ using a classification method (E). The subsequent regression is constrained by assuming that equal distance from the boundary should have approximately equal output value, in addition to considering the input combinations. This structure-augmented prediction gives an $R^2$ of 0.96 (F). Direct regression (bottom row), which directly maps inputs to the output, gives an $R^2$ of 0.81 (G).

On the full landscape, a key structure is the survival boundary by thresholding the landscape (Fig 1B and 1C). We learn this structure in two steps. First, we convert the density to binary classes by considering how well the plasmid-carrying population $S^1$ survives: if the density is greater than 0.2, it survives well; else, it survives poorly. We then learn the boundary using a support vector classifier (SVC). The learned boundary is denoted by the solid black line on Fig 1C. The figure also shows that an equal distance from the boundary corresponds to approximately equal values of the density of $S^1$, as evident in the approximately constant distance between neighboring contour lines. As such, the boundary represents an approximate lower-dimension (2D) signature of the overall response landscape (3D). SVC is suitable for the estimated distance assignment, since it returns a function for the learned boundary that can be used to calculate the distance (see Methods). Indeed, this distance is strongly correlated with the real $S^1$ density (Fig 1C, right panel).

Given sparsely yet sufficiently sampled data, the boundary can still be approximately reconstructed directly from the training data (Fig 1D and 1E). In this example, 10 data points suffice

for the boundary learning. The key to our method is to impose the learned boundary onto the subsequent regression analysis, as a constraint. Specifically, the regression model support vector regression (SVR) takes in three inputs, glucose and Lin concentrations, as well as the assigned distance of each training data from the learned classification boundary. To account for the possibility that the learned structure does not reflect the true data structure, especially when training data is sparse, in the regression learning step, we let the algorithm choose a weight from 0.01, 0.1, and 1 for this extra feature (see Methods). We find that our soft-constraint regression method is superior in the same prediction task, compared with direct regression (SVR alone) (Fig 1F and 1G). Specifically, after learning on the same 10 training data, SAR gives a prediction power of $R^2 = 0.96$, while the direct regression only has a prediction power of $R^2 = 0.81$. To test the robustness of this improvement, we train both methods on 40 different sets of 10 training data and test on the rest of the data, then compare the 40 $R^2$ value (S1B Fig). Indeed, our method performs statistically better than the simple regression (S1C Fig). To test whether the advantage of SAR is not limited to one training size, we increase the training data to 50, 10 at a time, and carry out the same statistical tests on the four new groups. SAR consistently outperforms direct regression in all cases (S1C Fig).

## Application of structure-augmented regression on simulation data

To examine the robustness of our method, we simulated dynamics of two biological systems that exhibit more complex response landscapes.

We first considered the same simulated community of $S^1$ and $S^0$ (Fig 1) using a different set of model parameters (Methods). The final density landscape of $S^1$ now has a much sharper transition (Fig 2B). While the required training data size increases, starting from 20 training data, SAR reaches a median $R^2$ of 0.7 and above, and it consistently outperforms direct regression (S2D Fig), which, given 20 training data, leads to a $R^2$ of 0.3. SAR captures the sharp transition at the bottom left of the landscape, while SVR alone misses this structure, resulting in a poor performance, even when using 50 data points (Fig 2C). Moreover, when the training dataset is small, the same method can lead to different prediction powers after learning from data of different distributions, as some distributions capture the overall dynamics and landscape better than others. However, SAR consistently shows a smaller variance in its performance across all training dataset sizes, thus exhibits a stronger reliability (S2D Fig). After training on 50 data points, SVR reaches a median $R^2$ of 0.5, with quartile one and three spans across $R^2$ of 0.25 and 0.85, while SAR reaches a median $R^2$ of 0.75 with quartile one and three spans across $R^2$ of 0.7 and 0.85, a much smaller variance. This smaller variance is again a result of the soft structural constraint imposed on regression.

We next considered a more complex community of two bacterial species transferring one plasmid (Methods). This community has four different populations ($S_1^1$, $S_1^0$, $S_2^1$, $S_2^0$), with each species either carrying or not carrying the plasmid (Fig 2D). Under the modulation of glucose and Lin, the density of final plasmid-carrying population shows an irregular structure with two underlying boundaries (Fig 2E). In this scenario, using the same framework, SAR learns a larger curved boundary that captures the two boundaries in one. As a result, SAR also consistently outperforms the direct regression: in terms of average $R^2$ and its variance when applied to different training data sets (S2E Fig). Again, as shown in Fig 2F, the SAR captures the irregular landscape, but SVR does not.

The key behind the better performance of SAR is the use of the structural constraint. Thus, the response landscape must have a sufficiently clear structure that can be deduced and applied for regression. As an illustration, we examine a fully random synthetic landscape as a negative

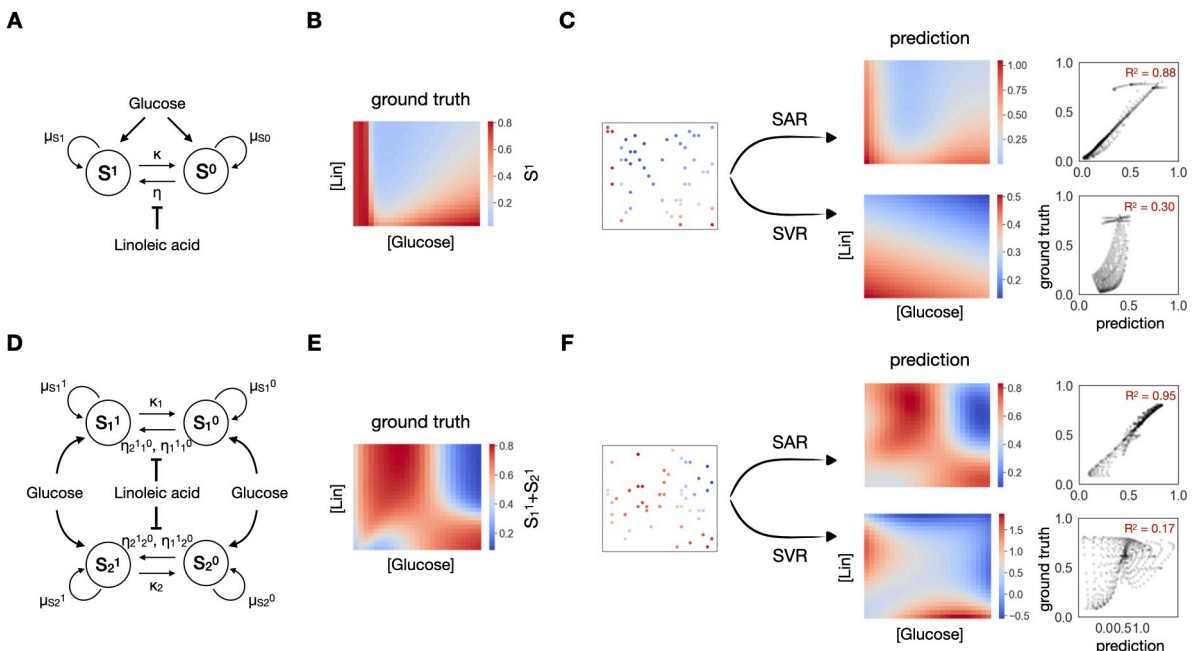

SAR: structure-augmented regression     SVR: support vector regression

**Fig 2. Better performance on simulation data of higher complexity.** A. Another community with one species transferring one plasmid, under modulation of a nutrient, glucose and an inhibitor, linoleic acid (Lin). B. Simulated response of final $S^1$ density in response to changing [glucose] and [Lin]. It shows a sharper transition at bottom left on the full simulated landscape. C. Prediction of the landscape using 50 data points with SAR (top) or SVR (bottom). SAR could capture the sharp transition, shown on the predicted heatmap and reach a $R^2$ of 0.88, while SVR alone fails to do so, reaching a $R^2$ of 0.30. D. A community with two species transferring one plasmid, resulting in four populations, with two carrying and two not carrying the plasmid. E. Simulated response of final $S_1^1 + S_2^1$ density in response to changing [glucose] and [Lin]. It shows a complicated landscape with two boundaries. F. Prediction of the landscape using 50 data points with SAR (top) or SVR (bottom). SAR could capture both boundaries, shown on the predicted heatmap and reach a $R^2$ of 0.95, while SVR alone fails to do so, reaching a $R^2$ of 0.17.

control (S2A Fig). SAR and SVR shows equally poor performance in learning the landscape (S2C Fig).

## SAR improves prediction on experimental data

We next examined the performance of SAR, in contrast to direct regression, on different types of experimental data. As a first example, we considered the response of a β-lactam-resistant *E. coli* population ($S^R$) in response to 100 combinations of a β-lactam antibiotic and a beta-lactamase inhibitor, created by 10 different concentrations of each drug (Fig 3A, See Methods for experimental details). The bacteria are resistant due to the expression of a beta-lactamase (Bla) that can degrade the antibiotic [50,51]; at a sufficiently high concentration, the Bla inhibitor inactivates Bla, thus sensitizing the bacteria to the antibiotic [52].

We measured the response of $S^R$ to the 100 combinations of β-lactam and Bla inhibitor concentrations (Fig 3B). Note that while the overall dynamics changes mostly along the vertical axis, i.e., the inhibitor concentration, there is a sharp transition between the survival and the death of $S^R$ as the Bla inhibitor increases. When applied to 30 points, SAR is able to predict the landscape with an averaged $R^2$ of 0.75, much better than the $R^2$ of 0.65 by direct regression as it captures this sharp transition (Figs 3C and S3A).

As another example, we considered a mixed community consisting of approximately equal fractions of $S^R$ and its sensitive counterpart $S^S$, the same strain without expressing Bla (Fig

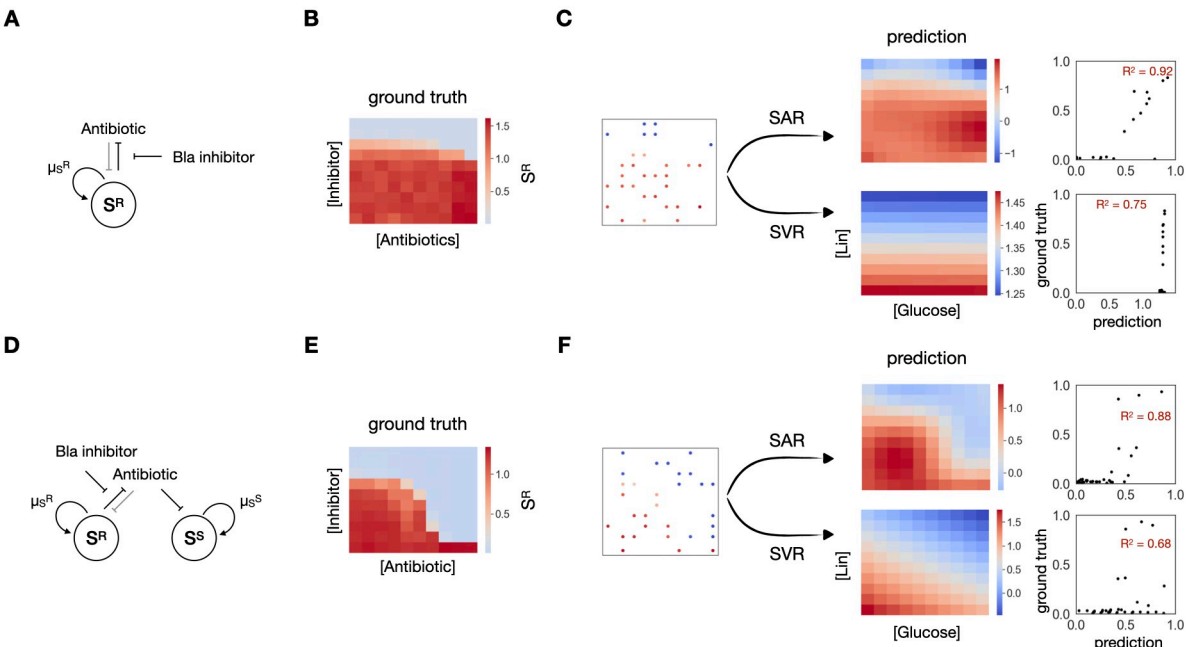

**Fig 3. Structural-augmented regression outperforms on 2D experimental data.** A. A β-lactam resistant *E. coli* community, under modulation of an antibiotics and a Bla inhibitor. B. Experimental results of final $S^R$ density in response to changing antibiotics and inhibitor concentrations. The response exhibits a sharp transition as the concentration of the inhibitor changes. C. Prediction of the landscape using 30 data points with SAR (top) or SVR (bottom). SAR could capture the sharp transition, shown on the predicted heatmap and reach a $R^2$ of 0.92, while SVR alone fails to do so, reaching a $R^2$ of 0.75. D. A mixed community consisting of approximately equal fractions of the resistant $S^R$ and the sensitive $S^S$ populations, under modulation of an antibiotics and a Bla inhibitor. E. Experimental results of final $S^R$ density in response to changing antibiotics and inhibitor concentrations. The response exhibits a slightly more complex landscape. F. Prediction of the landscape using 30 data points with SAR (top) or SVR (bottom). SAR could capture the complex landscape better, shown on the predicted heatmap and reach a $R^2$ of 0.88, than SVR alone, which reaches a $R^2$ of 0.68.

3D). In the mixture, $S^S$ can benefit from antibiotic degradation mediated by $S^R$. The two populations also interact due to competition for nutrient. In response to the same 100 combinations of the antibiotic and the Bla inhibitor, the mixture generates slightly more complex response landscape (Fig 3E). Again, when applied to 30 points sampled from this landscape, SAR outperforms SVR (Figs 3F and S3B), and it consistently does so when we increase the training data up to 50 points.

The benefit of imposing the structural constraint is not limited to the use of SVR for regression. We applied SAR using other regression methods, including polynomial regression, kNN and random forest to the two experimental systems and several more (S3 and S4 Figs). In each case, the use of the structural constraint in SAR improves the overall prediction performance in comparison to the counterpart that does not apply the constraint. Likewise, we replaced the classification method with logistic regression classification, and used the returned logistic regression value as the distance value; our conclusion held (S5 Fig).

## SAR facilitates active learning

The three simulated cases have shown that while SAR consistently outperforms direct regression, an increase of the landscape complexity also implies an increase in required training data even for SAR. While this is generally true for all data-driven algorithms, we want to further

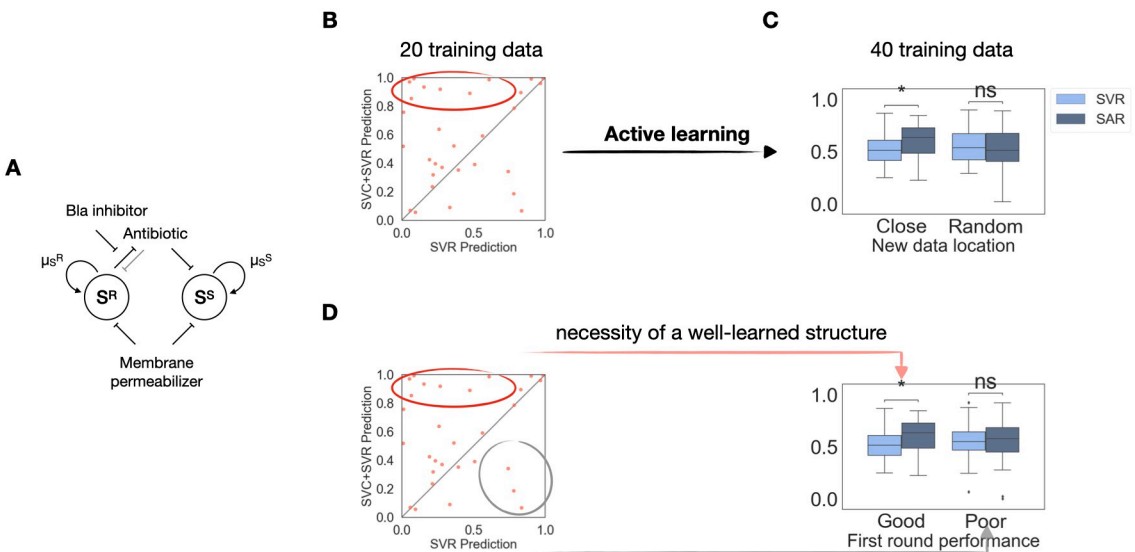

**Fig 4. Learned structure actively guides further experiments.** A. A mixed *E. coli* community consisting of approximately equal fractions of the resistant $S^R$ and the sensitive $S^S$ populations, under modulation of three drugs: a β-lactam antibiotic, a Bla inhibitor and a membrane permeabilizer. B. $R^2$ comparison after the first round of learning. Each dot represents performance of the two methods on one specific training set. The x-axis is the $R^2$ value of the simple regression prediction; the y-axis is the $R^2$ value of the SAR prediction. Majority of the scatter points aggregates above the diagonal line, showing that SAR outperforms the simple regression. C. Active learning needs to take advantage of the learned structure to work. Naive active learning, sampling 20 new points without taking advantage of the learned structure, does not outperform simple regression, as shown by the pair of bar plots on the right. While sampling 20 new points around the best learned boundary, indicated by the dots in the red circle, significantly improves the prediction accuracy. D. A well-learned structure is necessary to assist active learning. When the sampled data are based on the worst learned structures in the second round of data generation, indicated by the dots in the grey circle, SAR does not improve the prediction accuracy. p-value annotation legend: ns: $0.05 < p < = 1.0$; *: $0.01 < p < = 0.05$; **: $0.001 < p < = 0.01$; ***: $0.0001 < p < = 0.001$; ****: $p < = 0.0001$.

exploit the boundary information to generate high prediction accuracy with a minimal amount of data in systems with complicated landscape as well. So far, we have only applied SAR on 2D experimental systems, which generally show relatively simple landscapes. However, simple landscapes are not guaranteed in higher-dimensional systems. While exhaustively generating 100 different experimental conditions for a 2D system was doable, generating 1000 different conditions for a single 3D system is nontrivial. Comprehensive exploration of higher-dimensional response landscapes will be even more challenging or experimentally prohibitive. Therefore, further reducing the need for data for higher-dimensional systems, while maintaining high accuracy in predictions, is ideal.

To this end, we designed a 3D system based on the same β-lactam resistant community in Fig 3. In addition to the two drugs mentioned above, we apply an additional third drug, membrane permeabilizer to destabilize the membrane structure of the bacteria (Fig 4A). We then used this community to exploit the property of the boundary information. Specifically, we investigated whether the learned boundary can be used to guide the next round of experiments in a closed-loop ML-guided experimental design.

The general scheme of such ML-augmented guidance is called "active learning." Being "active" means ML actively takes part in the investigation process instead of merely serving as a passive prediction tool [53–55]. Our workflow contains two steps. After carrying out the first round of experiments, we first use SAR to learn from this round of experimental results as usual. The extra step is that it will now suggest another round of experiments that could

generate more informative data than randomly selected new ones. The goal of the active learning approach is to gather most information with minimal experiments. This approach has already been shown to reduce the total amount of experimental data needed to reach good prediction accuracies in various areas, especially in drug optimization and design [56,57].

Since we have already shown that learned boundary information is valuable in reducing the amount of data needed for a good prediction accuracy, we hypothesized that using this boundary to guide the active learning process could combine the two methods' advantages. We demonstrate the incorporation of SVR into active learning on the new β-lactam resistance experimental data.

For the first round of experiments, we randomly generated 25 three-drug combinations for the same initial bacterial community. We then let SAR learn from 20 random datapoints and tested its performance on the rest 5 datapoints. Since SAR can learn a different data structure from each training set, we applied SAR on 30 different training sets of 20 datapoints to ensure some learned structures are more representative of the true landscape. We then introduced the active learning by selecting 20 new datapoints to generate around the top learned boundaries. The quality of the boundary is evaluated by the improvement of the final regression prediction accuracy (Fig 4B). To test the prediction accuracy for the overall landscape, for the next round of experiments, we exhaustively generated all 64 combinations, 4 variations for each of the three conditions, but we only let the pipeline learn on the 40 datapoints, 20 from the previous experiments and 20 newly selected ones. After relearning, the pipeline is tested on 24 datapoints (see S6A Fig for implementation).

Since as the total training data increases, the prediction accuracy generally improves, we first evaluate whether taking advantage of the learned structure from the first iteration is necessary for accuracy improvement in the second round of learning. For this evaluation, we randomly generated 20 new datapoints without utilizing the knowledge of the learned boundary. As Fig 4C shows, this naïve active learning does not enable SAR to outperform the simple regression. However, if we sampled new data close to the top learned boundaries, our method significantly improves the prediction accuracy. Therefore, indeed, the improvement of the prediction performance is not simply due to the increase of the training data size.

We then verify that a well-learned structure in the first SAR step is also necessary in this active learning pipeline. To do so, instead of sampling new data closest to the best learned boundaries, we sampled the same amount of new data closest to the worst learned ones, i.e., the ones that result in the poorest SAR performances in the first round of learning (Fig 4D). As expected, sampling data based on these boundaries does not improve the prediction accuracy either. Therefore, both a well-learned structure from SAR and taking advantage of it in active learning are necessary.

Lastly, given the importance of a well-learned structure in the first step, we ask whether in the second round, further refining structure information is important also. We tested this by considering three selection schemes (S6B Fig). For each learned boundary, the first scheme is to sample new data around the learned boundary. The goal of this selection is to further refine the boundary accuracy during the second iteration of learning. The second option is to sample the data away from the learned boundary, so that the data will be highly segregated. The rationale of this choice is to take more advantage of the regression power by letting it learn points of more diverse quantitative values. The third scheme is to combine the advantage of classification and regression by picking half of the new experimental points close to the boundary and half of the new experimental points away from the boundary. Across the three different sampling schemes, the first showed significant improvement, where further structure refinement is the priority in the new data sampling (S6C Fig). Therefore, all three tests tell us that structure

information is essential to consider throughout active learning, which is a natural extension of our structural-augmented regression method.

## Structure-augmented regression is universally applicable to higher-dimensional systems

In combination therapies, up to eight drugs are being utilized [39] and in bacterial growth engineering, up to 10 nutrients are being tuned simultaneously [13]. To merely vary three concentrations for each of the 10 nutrients, there are $3^{10}$, 59,049 possible total combinations. Thus, a model that can learn well on few training data is especially valuable in this scenario.

To demonstrate that our method can be generalized to higher-dimensional systems, we analyzed three extra experimental datasets from different applications. First, we applied our method to an *E. coli* 3-sensor platform that could sense pH, thiosulfate (THS) and tetrathionate (TTR) (Fig 5A). This *E. coli* population contains three circuits, each senses one chemical respectively (see Methods). Specifically, upon sensing THS, *E. coli* would emit CFP fluorescence; it would also emit YFP fluorescence upon sensing TTR and mCherry fluorescence upon sensing pH. The concentration of each chemical determines the intensity of each corresponding fluorescence signal. In total, there are 16 concentration combinations in this dataset. After training on 10 combinations and testing on 6 combinations for THS sensing for 30 times, SVR reaches a median $R^2$ of 0.63, with quartile one and three spans across $R^2$ of 0.33 and 0.73, while SAR reaches a median $R^2$ of 0.70 with quartile one and three spans across $R^2$ of 0.52 and 0.83, a much smaller variance (Fig 5B).

We then applied our method to two published datasets. The first one is a multiple cancer drug combination treatment experimental dataset on lung cancer cell-line 786-O [58]. This dataset contains 50 7-drug combinations and their effects on the cancer cell-line (Fig 5C). We trained on 10, 20 or 30 different combinations and tested on the rest. While learning on 10 data points is not sufficient for either algorithm to excel, SAR outperforms on the 7-drug combination starting from learning on 20 data points. When training on only 20 data, SAR reached a median $R^2$ of 0.7, with the 1st and 3rd quantile spanning 0.6 to 0.8, while direct regression alone reached a median $R^2$ of 0.6, with the 1st and 3rd quantile spanning 0.3 to 0.7, a much wider variance than SAR (Fig 5D). The same dataset also contains some 4-drug combination tests, each has 25 combinations in total (S7C and S7D Fig). When training on 20 data, SAR performs well as well, reaching a median $R^2$ of 0.9, with the 1st and 3rd quantile spanning 0.7 to 1.0, while regression alone only reached a median $R^2$ of 0.7, with the 1st and 3rd quantile spanning only 0.2 to 0.8 (S7C and S7D Fig). We then further applied our method to a system of even higher dimension from a study of 10 common nutrients' effect on the growth of the bioplastic producing bacteria, *Cupriavidus necator* H16 [13] (Fig 5E). This dataset contains 64 combinations in total, so we trained on 10, 30 or 50 data and tested on the rest. Likely due to the high dimensionality of these two datasets, learning on 10 data was not sufficient, but SAR significantly outperformed regression alone when learning on 30 or 50 data (Fig 5F). Specifically, after learning on 50 data points, while regression could only reach a median $R^2$ of 0.2, SAR reached a median $R^2$ of 0.55.

## Discussion

ML has been recognized as an effective tool to predict the behavior of biological systems without thorough mechanistic understanding. Such predictions are valuable in engineering system behaviors and in aiding further explorations for deeper understanding of these systems. However, biological experiments and clinical applications usually do not generate enough data for ML methods to reach satisfying performance. This issue is exacerbated when the underlying

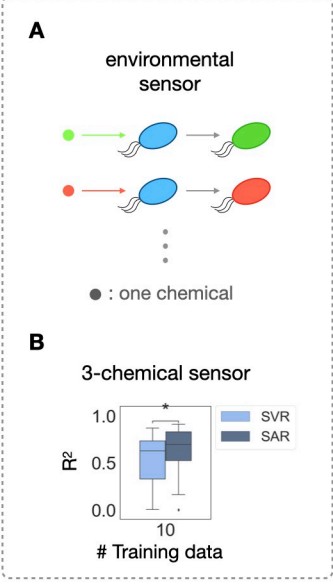
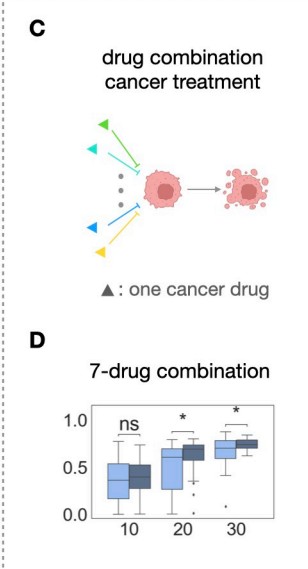
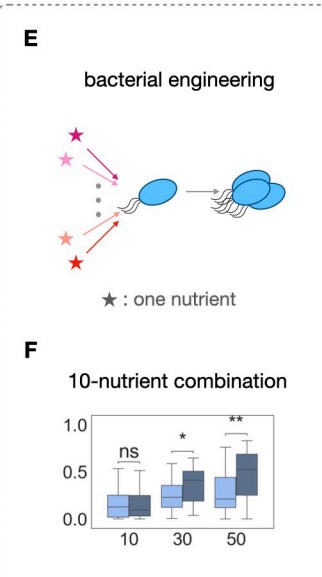

**Fig 5. Structural-augmented regression for higher-dimensional data prediction.** A. A 3-chemical *E. coli* sensor. Each chemical induces the bacteria to emit one type of fluorescence. For the ML pipelines, the input of each instance is the concentration combination of all the chemicals and the output is the fluorescence intensity. B. Statistical comparisons of regression and SAR. Given 16 different combinations in total, 10 different results are used to train both methods, with the rest of the data being the testing set. C. A 7-drug combination cancer treatment. Each input is the dose combination of all the drugs and the output is the final cell density (created with BioRender.com). D. Statistical comparisons of regression and SAR. Given 50 different combinations in total, 10, 20 and 30 different results are used to train both methods, with the rest of the data being the testing set. E. A 10-nutrient combination bacterial growth investigation. Each input is the concentration combination of all the nutrients and the output is the final cell density. F. Statistical comparisons of regression and structure-augmented regression. Given 64 different combinations in total, 10, 30 and 50 different results are used to train both methods, with the rest of the data as the testing set. p-value annotation legend: ns: $0.5 < p < = 1.0$; *: $0.01 < p < = 0.05$; **: $0.001 < p < = 0.01$; ***: $0.0001 < p < = 0.001$.

landscape of the system is complicated, which can be the case when the amount of input factors is large. Past studies have attempted to resolve this issue by exploiting existing ML methods via combining them in ensemble [33,46,47]. Here we provide a different approach by exploiting the data itself instead of the method. The key difference of our approach is to first extract an intrinsic property, the structural feature of the system, from the data itself. This structural feature is then used to augment the downstream analysis, regardless of the algorithms used. We envision that in the future, these two types of approaches can be used together. For instance, one can continue to use ensemble predictions, after imposing the structural constraint.

Like all regression ML methods, our method, SAR, first takes in a set of training instances, each consisting of the measurements of interest. Then the method provides a predictive model that maps the input to the desired output. The difference is that, during the training process, instead of directly learning the quantitative response from input measurements, the method first estimates the landscape from this set of training data, then learns the response from both input measurements and the estimated landscape. The landscape estimation takes advantage of classification algorithms that can draw a boundary between some critical transition of the system, which no quantitative learning algorithms have utilized before. Combining these two types of learning methods, SAR assumes that a biological system shows structured responses to a set of inputs, a feature that is common. Therefore, the essence of this strategy lies in our

ability to take advantage of this feature. To this end, we demonstrated that SAR can capture either one (Figs 1B and 2B) or two boundaries (Fig 2E). However, there is no clear way of knowing how many boundaries are necessary for a satisfying prediction power beforehand, especially in high-dimensional systems where data visualization itself is challenging. Therefore, determining critical transition boundaries will involve some trial and error based on data distribution. Note that the training process remains the same, as SAR will automatically learn the multiple boundaries in the original landscape.

Our method can be extended to biological systems where there are more than two phenotypes. Specifically, for applications with n classes (n>2), one can apply the standard one-against-all or one-against-one strategies [59,60] in the classification step. Specifically, to use the one-against-all strategy, one first trains n-1 one-against-all binary support vector classifiers and get n-1 boundaries. One then calculates the distance between each point and the boundaries as the calculated distances. Essentially, this provides n-1 new features for each point that capture the overall system response landscape. Another common classification method applicable to n classes is to train n(n-1) one-against-one binary support vector classifiers for each pair of classes and get n(n-1) boundaries. Likewise, one would then calculate each point's distance from all n(n-1) boundaries. This leads to n(n-1) new features.

Our method is complementary to other established methods for guiding experimental design or data analysis. Specifically, Design of Experiments (DOE) is one such method. During DOE, for each factor, a specific set of values to be tested are first determined from some previous screening or available literature data. These values are combined in a (full or partial) factorial manner for experimental data generations [61]. With our method, we could randomly select the values for each factor and then combine them in a random instead of in a factorial manner. Therefore, to use these two methods together, one can apply our method to the data selected from the DOE. Likewise, Bayesian optimization also estimates a landscape while optimizing a specific objective function. The key difference is that our method does not specifically aim to optimize any function. Instead, our method aims to estimate the overall landscape. To combine the advantages of these two methods, one can first use our method to learn the structural information as an additional input for the Bayesian optimization method.

We have demonstrated our method can be used in a range of applications where multiple factors affect system outputs, such as gene expression and metabolic pathway yields. As the number of factors increases, this method could become particularly useful as the exhaustive search will require an exponentially growing total number of combinations. However, if the system has no distinctive structure to be derived from the data, this method would not be applicable. Similarly, if certain landscape is so complex that no simple underlying structure could capture its major trend, this method would not be helpful either. However, it seems to be a common, both from literature and from our experimental data, for biological systems to respond in a structured manner to structured perturbations. Another advantage of our method is that, to further aid high-dimensional prediction, we can exploit the information from the learned landscape more by combining SAR with active learning. Specifically, we have demonstrated that this combination works the best when active learning suggests more data to collect around the previously learned boundary, which further takes advantage of a well-estimated landscape. As has been suggested recently that ML in biological applications will be particularly effective when combined with automation [25], we envision that combining the active learning version of SAR with automation will help to achieve the full potential of ML application in biological data.

## Methods

### Modeling

**One-species, one plasmid community.** For the one bacterial community that carries one plasmid, we simulated a community of two populations, $S^1$ represents the plasmid-carrying population, and $S^0$ represents a plasmid-free population. $S^0$ acquires the plasmid through conjugation at rate $\eta$, becoming $S^1$. $S^1$ reverts to $S^0$ through plasmid loss at rate $\kappa$. The conjugation rate $\eta$ is modulated by a conjugation inhibitor, linoleic acid (Lin in heatmap, $i$ in equations). The growth rates of both populations are modulated by a common nutrient, glucose ($g$).

Our model is as follows:

$$\frac{dS^1}{dt} = \left( \left( \mu_{S^1_{max}} - \mu_{S^1_{min}} \right) \left( \frac{g^{n_{s1}}}{g^{n_{s1}} + K_1^{n_{s1}}} \right) + \mu_{S^1_{min}} \right) S^1 (1 - S^1 - S^0) - \kappa S^1 + \left( (\eta_{max} - \eta_{min}) \left( \frac{K_i^{n_i}}{i^{n_i} + K_i^{n_i}} \right) + \eta_{min} \right) S^1 S^0 - D S^1 \quad (1)$$

$$\frac{dS^0}{dt} = \left( \left( \mu_{S^0_{max}} - \mu_{S^0_{min}} \right) \left( \frac{g^{n_{s0}}}{g^{n_{s0}} + K_0^{n_{s0}}} \right) + \mu_{S^0_{min}} \right) S^0 (1 - S^1 - S^0) + \kappa S^1 - \left( (\eta_{max} - \eta_{min}) \left( \frac{K_i^{n_i}}{i^{n_i} + K_i^{n_i}} \right) + \eta_{min} \right) S^1 S^0 - D S^0 \quad (2)$$

This model assumes that growth depends on nutrients according to Monod kinetics, modeled as Hill equations. The growth rate for $S_1$ can reach $\mu_{S^1_{max}}$ given sufficient nutrient ($g$). Likewise, the growth rate for $S_0$ can reach $\mu_{S^0_{max}}$ given sufficient $g$ as well. The model also assumes that the conjugation efficiency depends on conjugation inhibitor ($i$) in the same manner. The conjugation rate reaches $\eta_{max}$ when $I = 0$. Both populations are also subjected to dilution rate D.

For the simulated community in Fig 1, we used initial conditions of $S^1(0) = 0.1$, $S^2(0) = 0.1$, $5 < g < 50$, $5 < i < 50$. Parameters used for were: $\mu_{S^1_{max}} = 0.7$; $\mu_{S^1_{mix}} = 0.05$; $n_{S^1} = 0.1$; $K_1 = 40$; $\mu_{S^0_{max}} = 0.8$; $\mu_{S^0_{min}} = 0.05$; $n_{s0} = 0.5$; $K_0 = 10$; $\kappa = 10^{-3}$; $\eta_{max} = 10^{-1}$; $\eta_{min} = 10^{-5}$; $n_i = 1$; $K_i = 20$; D = 0.05. For the simulated community in Fig 2A, we used initial conditions of $S^1(0) = 0.1$, $S^2(0) = 0.1$, $5 < g < 50$, $5 < i < 50$. Parameters used for were: $\mu_{S^1_{max}} = 0.7$; $\mu_{S^1_{min}} = 0.05$; $n_{S^1} = 1$; $K_1 = 15$; $\mu_{S^0_{max}} = 0.8$; $\mu_{S^0_{min}} = 0.05$; $n_{s0} = 5$; $K_0 = 10$; $\kappa = 10^{-3}$; $\eta_{max} = 10^{-1}$; $\eta_{min} = 10^{-5}$; $n_i = 0.5$; $K_i = 2$; D = 0.05.

To understand the mathematical implications of these models in more detail, please see references [7,62,63].

**Two-species, one plasmid community.** For the community of two bacterial species $S_1$ and $S_2$, transferring one plasmid, we simulated a community of four subpopulations, $S_1^0$, $S_1^1$, $S_2^0$ and $S_2^1$. $S_1^0$ acquires the plasmid through conjugation at rate $\eta_1{}^1{}_1{}^0$ from $S_1^1$, and $\eta_2{}^1{}_1{}^0$ from $S_2^1$, becoming $S_1^1$. $S_1^1$ reverts to $S_1^0$ through plasmid loss at rate $\kappa_1$. Likewise, $S_2^0$ acquires the plasmid through conjugation at rate $\eta_1{}^1{}_2{}^0$ from $S_1^1$, and $\eta_2{}^1{}_2{}^0$ from $S_2^1$, becoming $S_1^1$. $S_2^1$ reverts to $S_2^0$ through plasmid loss at rate $\kappa_2$. All conjugation rates are modulated by a conjugation inhibitor, linoleic acid (Lin in heatmap, $i$ in equations). The growth rates of all four populations are modulated by a common nutrient, glucose ($g$).

Our model is as follows:

$$\frac{dS_1^1}{dt} = \left( \left( \mu_{S_1^{11}_{max}} - \mu_{S_1^{11}_{min}} \right) \left( \frac{g_1{}^{n_1}}{g_1{}^{n_1} + K_1{}^{n_1}} \right) + \mu_{S_1^{11}_{min}} \right) S_1^1 \left( 1 - S_1^0 - S_1^1 - S_2^0 - S_2^1 \right) - \kappa_1 S_1^1 +$$
$$\left( \left( \eta_{1^1 1^0 max} - \eta_{1^1 1^0 min} \right) \left( \frac{K_{i1}{}^{n_{i1}}}{i1^{n_{i1}} + K_{i1}{}^{n_{i1}}} \right) + \eta_{1^1 1^0 min} \right) S_1^1 S_1^0 + \quad (3)$$
$$\left( \left( \eta_{2^1 1^0 max} - \eta_{2^1 1^0 min} \right) \left( \frac{K_{i2}{}^{n_{i2}}}{i2^{n_{i2}} + K_{i2}{}^{n_{i2}}} \right) + \eta_{1^1 1^0 min} \right) S_2^1 S_1^0 - D S_1^1$$

$$\frac{dS_1^0}{dt} = \left(\left(\mu_{S_{max}^{10}} - \mu_{S_{min}^{10}}\right)\left(\frac{g_2^{\,n_2}}{g_2^{\,n_2} + K_2^{\,n_2}}\right) + \mu_{S_{min}^{10}}\right)S_1^0\left(1 - S_1^0 - S_1^1 - S_2^0 - S_2^1\right) +$$
$$\kappa_1 S_1^1 - \left(\left(\eta_{1^1 1^0\,max} - \eta_{1^1 1^0\,min}\right)\left(\frac{K_{i1}^{\,n_{i1}}}{i1^{n_{i1}} + K_{i1}^{\,n_{i1}}}\right) + \eta_{1^1 1^0\,min}\right)S_1^1 S_1^0 - \qquad (4)$$
$$\left(\left(\eta_{2^1 1^0\,max} - \eta_{2^1 1^0\,min}\right)\left(\frac{K_{i2}^{\,n_{i2}}}{i2^{n_{i2}} + K_{i2}^{\,n_{i2}}}\right) + \eta_{1^1 1^0\,min}\right)S_2^1 S_1^0 - D S_1^0$$

$$\frac{dS_2^1}{dt} = \left(\left(\mu_{S_{max}^{21}} - \mu_{S_{min}^{21}}\right)\left(\frac{g_3^{\,n_3}}{g_3^{\,n_3} + K_3^{\,n_3}}\right) + \mu_{S_{min}^{21}}\right)S_2^1\left(1 - S_1^0 - S_1^1 - S_2^0 - S_2^1\right) -$$
$$\kappa_2 S_2^1 + \left(\left(\eta_{1^1 2^0\,max} - \eta_{1^1 2^0\,min}\right)\left(\frac{K_{i3}^{\,n_{i3}}}{i3^{n_{i3}} + K_{i3}^{\,n_{i3}}}\right) + \eta_{1^1 2^0\,min}\right)S_1^1 S_2^0 + \qquad (5)$$
$$\left(\left(\eta_{2^1 2^0\,max} - \eta_{2^1 2^0\,min}\right)\left(\frac{K_{i4}^{\,n_{i4}}}{i4^{n_{i4}} + K_{i4}^{\,n_{i4}}}\right) + \eta_{1^1 2^0\,min}\right)S_2^1 S_2^0 - D S_2^1$$

$$\frac{dS_2^0}{dt} = \left(\left(\mu_{S_{max}^{20}} - \mu_{S_{min}^{20}}\right)\left(\frac{g_4^{\,n_4}}{g_4^{\,n_4} + K_4^{\,n_4}}\right) + \mu_{S_{min}^{20}}\right)S_2^0\left(1 - S_1^0 - S_1^1 - S_2^0 - S_2^1\right) +$$
$$\kappa_2 S_2^1 - \left(\left(\eta_{1^1 2^0\,max} - \eta_{1^1 2^0\,min}\right)\left(\frac{K_{i3}^{\,n_{i3}}}{i3^{n_{i3}} + K_{i3}^{\,n_{i3}}}\right) + \eta_{1^1 2^0\,min}\right)S_1^1 S_2^0 - \qquad (6)$$
$$\left(\left(\eta_{2^1 2^0\,max} - \eta_{2^1 2^0\,min}\right)\left(\frac{K_{i4}^{\,n_{i4}}}{i4^{n_{i4}} + K_{i4}^{\,n_{i4}}}\right) + \eta_{1^1 2^0\,min}\right)S_2^1 S_2^0 - D S_2^0$$

This model assumes that growth depends on nutrients according to Monod kinetics, modeled as hill equations. The growth rate for $S_1^1$ can reach $\mu_{S_{max}^{11}}$ given sufficient glucose ($g$). The same applies to all other three populations. The model also assumes that the conjugation efficiency depends on conjugation inhibitor ($i$) in the same manner. The conjugation rate between $S_1^1$ and $S_1^0$, $\eta_{1^1 1^0}$ reaches $\eta_{1^1 1^0\,max}$ when $i = 0$. All four populations are also subjected to dilution rate D.

For the simulated community in Fig 2D, we used initial conditions of $S_1^1(0) = 0.1$, $S_1^0(0) = 0.1$, $S_2^1(0) = 0.1$, $S_2^0(0) = 0.1$, $5 < g < 50$, $5 < i < 50$. Parameters used for were: $\mu_{S_{max}^{11}} = 0.6$; $\mu_{S_{min}^{11}} = 0.05$; $\mu_{S_{max}^{10}} = 0.9$; $\mu_{S_{min}^{10}} = 0.00$; $\mu_{S_{max}^{21}} = 0.5$; $\mu_{S_{min}^{21}} = 0.05$; $\mu_{S_{max}^{20}} = 0.85$; $\mu_{S_{min}^{20}} = 0.00$; $n_1 = 2$; $K_1 = 25$; $n_2 = 1$; $K_2 = 40$; $n_3 = 2$; $K_3 = 20$; $n_4 = 2$; $K_4 = 35$; $\eta_{1^1 1^0\,max} = 0.25$; $\eta_{1^1 1^0\,min} = 10^{-4}$; $\eta_{2^1 1^0\,max} = 0.25$; $\eta_{2^1 1^0\,min} = 10^{-4}$; $\eta_{1^1 2^0\,max} = 10^{-1}$; $\eta_{1^1 2^0\,min} = 10^{-4}$; $\eta_{2^1 2^0\,max} = 10^{-1}$; $\eta_{2^1 2^0\,min} = 10^{-4}$; $K_{i1} = 1$; $n_{i1} = 5$; $K_{i2} = 1$; $n_{i2} = 5$; $K_{i3} = 3$; $n_{i3} = 10$; $K_{i4} = 3$; $n_{i4} = 10$; D = 0.05.

## Structure-augmented regression

The structure-augmented regression is a regression method that also takes advantage of the classification method. The classification, support vector classification (SVC), is done first to aid the regression. The choice of the regression method is flexible, which includes support vector regression (SVR), linear regression, polynomial regression, k-nearest neighbor (KNN) and random forest (RF). In the two-step regression pipeline, we utilized the distance function returned by SVC as the additional distance input parameter for the following regression function.

Specifically, the labels for the first classification task represent the two distinct phenotypes of the biological system subjecting to different perturbations. Under antibiotic combination

treatments or cancer drug combination treatments, the two distinct phenotypes would be cell survival versus death. In response to nutrient combinations, the two distinct phenotypes would be high cell yield versus low cell yield. In a biosensing application, the two phenotypes are responses that pass a certain acceptable threshold or not.

In the following regression task, the quantitative values to be predicted are the cell densities or sensory responses (i.e., fluorescence measurements) in different environments.

For the concreteness of the following explanation, we will use the drug combination treatment as an example where we explain the whole SAR working follow. However, this method is applicable to all systems mentioned in this paper and beyond, as long as the quantitative output of the biological system under various perturbation can be categorized under a reasonable criterion.

For the first classification step, the input data for SVC are the following:

$$\text{Label of cell survival versus death}: \mathbf{Y} = \left[y_1, \ldots, y_i, \ldots, y_n\right]. \tag{7}$$

$$\text{Tuning parameters}: \mathbf{p} = [p_1, \ldots, p_i, \ldots, p_n]. \tag{8}$$

In Eqs (7) and (8), n represents total number of datapoints, either from simulations or experiments, and each index represents one observation. Each $y_i$ takes values of 1 or −1, representing cell survival versus death. The list $\mathbf{p}$ contains the experimentally controlled conditions, under which the observations are obtained; and each $p_i$ is a vector of which each element represents a tunable condition. For a system with two tunable conditions, like the ones in Figs 1, 2 and 3, $p_i = (p_{i1}, p_{i2})$.

The outputs of SVC include both the predicted labels and a distance function:

$$\text{Predicted label of cell survival versus death}: \mathrm{Y_{pred}} = \left[y_{\mathrm{pred1}}, \ldots, y_{\mathrm{predi}}, \ldots, y_{\mathrm{predn}}\right]. \tag{9}$$

$$\text{The distance function}: f(\boldsymbol{p}) = \sum_i \alpha_i y_i K\langle p_i, p \rangle + \lambda_0, \tag{10}$$

$y_i$ and $p_i$ are input values for observation i. $\alpha_i$ is the weight of observation i, and $\lambda_0$ is the bias term. $K$ here represents a kernel function that transforms the input data into a new space where the transformed data is now linearly separable by SVC in that space. Both the parameters $\alpha_i$ and $\lambda_0$, and the kernel, as well as the kernel hyperparameters are optimized by the SVC algorithm. Specifically, we choose the best kernel from linear, polynomial and RBF kernels based on the performance of SVC. Likewise, the kernel hyperparameters, i.e., the degree and the coefficient term for the polynomial kernel, and the gamma term for both polynomial and RBF kernels are all optimized with cross-validation. We used python's scikit packages for all training and optimization.

The function $f$ returns a numerical value for each input $p$. For datapoints on the learned classification boundary from SVC, the numerical values are all optimized to be 0. SVC also locates a set of support vectors from the input vector list $\mathbf{p}$ that are closest to the boundary from both classes (1 and -1); $\alpha_i$ for these vectors and vectors on the boundary are non-zero, whereas $\alpha_i$ for all other vectors are zero.

For any unseen input $p$, since the numerical value $f(\boldsymbol{p})$ is the sum of dot product of $p$ and each support vector, the magnitude of the value reflects the distance between $p$ and the boundary in the kernel-transformed space. Note given that the optimal kernel $K$ might not be a linear kernel, a nonlinear kernel can return $f(\boldsymbol{p})$ values that are not the same for points that are the same distance away from the boundary in the original space. However, as $f(\boldsymbol{p})$ have a monotonical dependence on the distance to the boundary in the transformed space, we found

empirically, this monotonic relationship between $f(\boldsymbol{p})$ and distance in the transformed space could still assist the regression by providing useful information.

This magnitude $f(\boldsymbol{p})$ is the additional parameter for the following regression step.

For the regression step, the input for SVR or any other regression is:

$$\text{Cell density}: \mathbf{D} = [d_1, \ldots, d_i, \ldots, d_n]. \tag{11}$$

$$\text{A new set of input parameters}: \mathbf{p'} = [p'_1, \ldots, p'_i, \ldots, p'_n]. \tag{12}$$

In Eqs (5) and (6), n represents total number of datapoints like in Eqs (7) and (8), and each index represents one observation. Each $d_i$ takes a quantitative value of the cell density. The list $\mathbf{p'}$ now contains both the experimentally controlled conditions and the estimated structure information represented by the SVC-calculated numerical value of the experimental conditions. Specifically, each $p'_i$ is a vector that contains two parts. The first part is the tunable condition $p_i$ and the second is the estimated distance returned by $\mathbf{f}(p_i)$ from SVC. For a system with two tunable variables, $p'_i = (p_{i1}, p_{i2}, f(p'_i))$.

During the learning process, a soft constraint is applied to the additional feature from classification, $f(p'_i)$, by letting the regression method choose the optimal weight $w$ for it, along with other regression hyperparameters. The $w$ is one of the three values from [0.01, 0.1, 1]. The rationale is that this estimated structure might not be representative if the training data is sparce.

$$\text{The output of SVR is the predicted cell density}: \mathbf{D_{pred}} = \left[d_{pred1}, \ldots, d_{predi}, \ldots, d_{predn}\right]. \tag{13}$$

The hyperparameter tuning of both the classification and regression algorithms are done by random search in a 5-fold cross validation.

## Methods comparison

We first randomly split the ground truth into training and testing sets. The size of the training set is 10, 20, 30, 40 or 50, according to the total amount of available experimental data from each system. We then train the traditional regression model and our structure-augmented regression model on the same training set simultaneously. The two learned models will then be applied to the same testing set for prediction. The prediction accuracy is evaluated using $R^2$ measurements.

To test the significance of the method improvement, for each dataset, we carried out the same training and testing procedure 40 times on 40 different randomly split training and testing sets for each amount of training data. We then compare the pair of 40 $R^2$ values using Mann-Whitney test (S1 Fig).

## Application of SAR to active learning

SAR can be applied to the active learning framework in an iterative four-step process. We demonstrated this combined pipeline on a 3-drug combination treated antibiotic resistance dataset. After generating the initial experimental dataset, the combined pipeline consists of four steps, which is described in detail in the following paragraphs. The four steps in sequence are method comparison application to different training and testing sets, informative learned boundary selection, new training data selection and experimental generation, and model and boundary updating (S5A Fig). Iteration of these four steps can happen for multiple rounds if needed.

In our application, the specific procedure is the following: the initial experimental dataset consists of 25 datapoints, generated under 25 randomly picked drug combinations out of the 4*4*4 = 64 exhaustive combinations. The first step of the active learning is applying the method comparison pipeline on the initial experimental data to learn 30 different structures by training on 30 different randomly selected training sets. Specifically, for each learning, we randomly split the 25 datapoints into a training set of 20 points and a testing set of 5 points. Due to the difference in the 30 training dataset distributions, we get 30 different learned boundaries, as well as 30 pairs of prediction accuracies of the testing sets from the two methods, i.e., 30 pairs of $R^2$ of SAR, $R^2_{SAR}$, and of SVR, $R^2_{SVR}$. In the second step, we use the difference in each pair as a metric to measure the informativeness of the 30 learned boundaries. Specifically, higher $R^2_{SAR}$—$R^2_{SVR}$ values correspond to better learned boundaries. We only consider learned boundaries that give positive $R^2_{SAR}$—$R^2_{SVR}$ values in the following steps, as only these ones provide useful information. The third step is, for each selected boundary from the previous step, generating 20 more experimental training datapoints using the learned boundary and generating the rest of the 24 datapoints for testing. Lastly, the ML pipeline is applied on the data generated in both rounds. If needed, the number of iterations for active learning, can increase further. For our application, we did one round.

During the third step, we investigate three sampling strategies (S5B Fig). Given that SAR takes advantage of the underlying structures by identifying a key boundary, the first strategy is to further refine this boundary in the additional data generation step. We implement this strategy by picking the additional 20 data that are closest to the learned boundary in the previous round, 10 on each side of the boundary. Such selection can be determined by the $f(\boldsymbol{p})$ values, i.e., 10 points of the smallest positive $f(\boldsymbol{p})$ values and 10 points of the largest negative $f(\boldsymbol{p})$ values. The second strategy is to explore previously less explored sample space by selecting new data that are farthest away from the boundary. This is done by choosing 10 new points of the largest positive $f(\boldsymbol{p})$ values and 10 new points of the smallest negative $f(\boldsymbol{p})$ values. Given these points are farther apart in the sampling space than the points closest to the boundary, they provide more information for the regression step instead of the boundary estimation step. The third strategy is to combine these two strategies by sampling half of the new data closest to the boundary and the other half farthest from the boundary.

## Antibiotic-resistant community

All experiments are done on two *E. coli* strains, Top10F' or DA28102. Each strain contains either a high-copy-number or a moderate-copy-number plasmid expressing a β-lactamase. We created two types of communities, either clonal resistance communities or a mixture community with approximately half of the initial population being the sensitive background strain.

We treated the communities using combinations of an antibiotic and a β-lactamase inhibitor. The antibiotic is amoxicillin, and the inhibitor is either clavulanic acid (CLA) or sulbactam (SUL). For the three-drug combination, the additional drug is an out membrane permeabilizer, polymyxin B nonapeptide (PMBN).

## Growth conditions of the antibiotic-resistance community

To generate bacterial growth in a high-throughput manner, we used the following protocol. Frozen stocks were streaked on lysogeny broth (LB) agar plates, and individual colonies selected to inoculate growth media. Overnight cultures of strains (prepared separately for mixed cultures) were prepared in 2 mL of LB broth in 15 mL culture tubes (Olympus) with 1 mM IPTG and 50 μg/mL kanamycin for plasmid selection if applicable; tubes were shaken at 37°C for 16 h at 225 rpm. The OD (absorbance at 600 nm) for the overnight culture was taken

on a plate reader (Tecan Spark multimode microplate reader). To ensure a consistent initial cell number, cultures were diluted to 1 OD600 (assumed to be equivalent to $8 \times 10^8$ cells/mL). For mixed culture experiments, equal volumes of each population were mixed at this step. For all experiments, the resulting culture was further diluted 1:8 ($1 \times 10^8$ cells/mL). Cultures were then finally diluted 10-fold in 100 μL of media in a 384-well deep-well plate (Thermo Scientific) using a MANTIS liquid handler for an initial cell density of $1 \times 10^6$ cells/well. The media in each well, containing the appropriate amounts of amoxicillin and beta-lactamase inhibitor (clavulanic acid, or sulbactam) for a total of 100 different conditions, was prepared ahead of time in appropriate concentrations and dispensed using the MANTIS liquid handler. Three technical replicates (3 separate wells) were generated for each condition. The spatial position of all wells for each experiment was randomized across the plate to minimize plate effects. To minimize evaporation, the plate was loaded with the lid into the Tecan Spark microplate reader equipped with a lid lifter, and the chamber temperature was maintained at 30˚C. OD600 readings were taken every 10 min with periodic shaking (5 s orbital) for 24 h.

### *E. coli* sensor design

The pH, TTR, and THS sensor strains were created by modifying previously developed two-component systems [64–66]. Plasmids pKD279.8, pKD280.7, pKD236-4b, pKD237-3a-2, pKD238-1a, and pKD239-1g-2 were gifts from Jeffrey Tabor lab. The plasmids containing the regulator gene and fluorescence reporter gene (pKD239-1g-2, pKD237-3a-2, pKD279.8) were modified to each have a different fluorescent reporter gene, and the constitutively expressed mCherry gene was removed. The mCherry, YFP, and CFP gene sequences were obtained from plasmids previously developed in our lab. Plasmid modifications were performed using polymerase chain reaction (PCR) and Gibson Assembly cloning method and verified through whole plasmid sequencing by Primordium Labs [67]. The other plasmids were unmodified. The two plasmids for each sensor were co-transformed into an MG1655 *E. coli* strain via CaCl2 transformation. Colonies were cultured in chloramphenicol and spectinomycin overnight, then glycerol was added to create a freezer stock with 25% glycerol.

### Sensor experiment protocol

Freezer stocks were streaked onto LB agar plates with chloramphenicol and spectinomycin and grown overnight. Single colonies were picked and cultured overnight in 3 mL LB media with appropriate antibiotics. OD was measured, cells were pelleted by centrifugation, and the supernatant was removed. Cultures were resuspended in LB pH 5.79, LB pH 8.05, M9 with 0.04% glucose pH 5.73, or M9 with 0.04% glucose pH 7.25 each to a final OD of 0.45. Appropriate antibiotics, 1 mM IPTG, and 100 ng/uL aTc were added to all tubes. Cultures were distributed to 96-well plates (Corning) and 1 mM THS or 1 mM TTR was added to the appropriate wells. Mineral oil was added on top to prevent evaporation. OD and fluorescence intensity was measured every 5 minutes using a plate reader (Tecan).

## Supporting information

**S1 Fig. Training and testing procedures to compare pipeline performances.** A. The ground truth to demonstrate the flow of training and testing procedure. The x- and y- axis represent two environmental factors that can be tuned to control the growth of some biological system: in this case, glucose and linoleic acid (Lin). The heatmap color represents the final cell density of that system. B. The training and testing procedure. We first split the ground truth into training and testing sets. Note the small size of training set corresponds to the limited amount of experimental data usually available. We then train the traditional regression model (direct

regression pipeline) and our structure-augmented regression (SAR) model on the same training set simultaneously. The two learned models will then be applied to the same testing set for prediction. The traditional regression pipeline is represented by the light blue arrows and the structure-augmented regression pipeline is represented by the dark blue arrows throughout this figure and all other figures in the manuscript. The prediction accuracy is evaluated using $R^2$ measurements. Here we just show one set of 10 training data. The "X40" means we carry out the same training and testing procedure 40 times on 40 different training and testing set splits. We then compare the 40 pairs of $R^2$ using the Mann-Whitney statistical test. This gives us the pipeline performance comparison on one specific training data amount, highlighted in orange in C. C. For each of the 10, 20, 30, 40 or 50 specific amount of training data, we carry out the same type of pipeline comparison shown in B. This gives us a more comprehensive view of method performance across a wide range of training data availability. p-value annotation legend: ns: $0.05 < p < = 1.0$; *: $0.01 < p < = 0.05$; **: $0.001 < p < = 0.01$; ***: $0.0001 < p < = 0.001$; ****: $p < = 0.0001$.
(TIFF)

**S2 Fig. Structure is essential for method application.** A. Ground truth of a synthetic random landscape. B. One learning example of the random landscape. The learning algorithm struggles to learn any structure. C. The method comparison of the two methods. The performance evaluation is done in the same way as in S1 Fig. The two methods perform equally poorly on a landscape with no structure. D. Method comparison on the community of one species and one plasmid as in Fig 2A. Structure-augmented regression consistently outperforms starting from with a training data set of 20 points and shows less variance. E. Method comparison on the community of two species and one plasmid as in Fig 2D. Structure-augmented regression consistently outperforms starting from with a training data set of 20 points and shows less variance as well. p-value annotation legend: ns: $0.05 < p < = 1.0$; *: $0.01 < p < = 0.05$; **: $0.001 < p < = 0.01$; ***: $0.0001 < p < = 0.001$; ****: $p < = 0.0001$.
(TIFF)

**S3 Fig. Structure-augmented regression is a flexible pipeline.** Statistical tests of the regression and structure-augmented regression methods on the two communities in Fig 3. The first row corresponds to the example in Fig 3A–3C. The second row corresponds to the example in Fig 3D–3F. Here we include three different kinds of regression methods: SVR, polynomial regression and random forest regression. The layout of the two rows (A and B) are the same. The first panel from left: schematic of the community. The second panel: ground truth of final population density. The next three panels: method comparisons using flexible ML pipelines. We can see that structure-augmented regression consistently outperforms the regression itself for all these three regression methods. The improvement for the first dataset is not as obvious, since that the landscape itself in this sample is very simple. p-value annotation legend: ns: $0.05 < p < = 1.0$; *: $0.01 < p < = 0.05$; **: $0.001 < p < = 0.01$; ***: $0.0001 < p < = 0.001$; ****: $p < = 0.0001$.
(TIFF)

**S4 Fig. Additional application on experimental data demonstrates the generalizability of the regression method in the pipeline.** All experimental data here are still the final cell density of β-lactam resistance communities, under combination treatments. The communities are either DH5α or Top10F' *E.coli* cells, shown as labels on top of each ground truth panel. All combinations use two drugs: one antibiotic and one β-lactamase inhibitor. Each row represents application on one specific experimental result. The layout of each row is the following: Left panel: ground truth of final population density. The next four panels: method

comparisons using flexible ML pipeline that integrates various regression methods with SVC, including SVR, KNN, polynomial regression and random forest regression. We can see that when applying to landscapes of different types of structures, our method consistently improves the prediction accuracy. p-value annotation legend: ns: $0.05 < p <= 1.0$; *: $0.01 < p <= 0.05$; **: $0.001 < p <= 0.01$; ***: $0.0001 < p <= 0.001$; ****: $p <= 0.0001$.
(TIFF)

**S5 Fig. Additional application on experimental data demonstrates the flexibility of the classification method in the pipeline.** All experimental data here are the same as in S4 Fig (still the final cell density of β-lactam resistance communities, under combination treatments). The communities are either DH5α or Top10F' *E.coli* cells, shown as labels on top of each ground truth panel. All combinations use two drugs: one antibiotic and one β-lactamase inhibitor. Each row represents application on one specific experimental result. The layout of each row is the following: Left panel: ground truth of final population density. The next four panels: method comparisons using flexible ML pipeline that integrates various regression methods with logistic regression classifier, including SVR, KNN, polynomial regression and random forest regression. We can see that when applying to landscapes of different types of structures, our method with logistic regression classifier in the first step also consistently improves the prediction accuracy. p-value annotation legend: ns: $0.05 < p <= 1.0$; *: $0.01 < p <= 0.05$; **: $0.001 < p <= 0.01$; ***: $0.0001 < p <= 0.001$; ****: $p <= 0.0001$.
(TIFF)

**S6 Fig. Workflow that combines structure-augmented regression and active learning.** A. The four-step process of combining structure-augmented regression and active learning. The key is to utilize the learned structure in future rounds of experimental data generation after the first round of learning. After experimentally generating 20 data, the first step is to apply the pipeline on them as usual to learn 30 different structures by learning on 30 different training sets. The second step is to pick the top informative boundaries out of the 30 learned ones by final prediction accuracy improvement of SAR over SVR. The boundary that contributes to the most improvement is the most informative. The third step is to generate the next round of experimental data based on the best boundary. Fourth, the ML pipeline is applied on the data generated in both rounds. If needed, the number of iterations, n, can increase for further actively learn. To achieve statistically significant comparison, in our test, we pick the boundaries that contribute to an increase of $R^2$ value $> 0.1$, sample new data around each of these boundaries and relearn, then compare two sets of new $R^2$ values. B. Cartoon for three types of combination schemes. Assume that the dashed black line in the middle of the scatter plot is the learned boundary from the first round of learning, there are three combination schemes given this information. There are three schemes of the new data selection: refining boundary by selecting data around the boundary; exploiting regression by selecting data away from the boundary; combining the advantages of classification and regression by combing these two approaches. C. Second round of experiments needs to further refine learned structure. There are three schemes of the new data generation based on the best learned structure of the first round: refining boundary by selecting more data around the boundary; exploiting regression by selecting data away from the boundary; combining the advantages of these two strategies. Running the same algorithm on new data generated from these three strategies shows that only the first sampling scheme works well with active learning.
(TIFF)

**S7 Fig. More applications on high-dimensional drug combination treatment.** A. Illustration of applying four different cancer drugs for better treatment results (created with BioRender.

com). B. The seven drugs and their tested range for Fig 5A. (C, D.) Applications on two different 4-drug combination treatments of cancer cell-line 786-O. Both figures follow the same format. The top table contains the drug and dosage information; the bottom figure is the method comparison results. Both experiments generated 25 datapoints in total, so the method comparison only trained on either 10 or 20 data and tested on the rest. The structure-augmented regression consistently outperforms direct regression while being trained on 20 datapoints. (TIFF)

## Acknowledgments

We thank Rohan Maddamsetti for assistance in editing the manuscript.

## Author Contributions

**Conceptualization:** Yuanchi Ha, Lingchong You.

**Data curation:** Yuanchi Ha, Helena R. Ma, Katherine Duncker, Helen Z. Xu.

**Formal analysis:** Yuanchi Ha.

**Funding acquisition:** Lingchong You.

**Investigation:** Yuanchi Ha, Feilun Wu, Andrea Weiss, Lingchong You.

**Methodology:** Yuanchi Ha.

**Project administration:** Yuanchi Ha, Lingchong You.

**Resources:** Yuanchi Ha.

**Supervision:** Daniel Reker, Lingchong You.

**Validation:** Yuanchi Ha, Jia Lu, Max Golovsky.

**Visualization:** Yuanchi Ha.

**Writing – original draft:** Yuanchi Ha, Lingchong You.

**Writing – review & editing:** Yuanchi Ha, Helena R. Ma, Feilun Wu, Andrea Weiss, Daniel Reker, Lingchong You.

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
