## [Decision Letter · Decision Letter 0]

4 Feb 2024

Dear Dr. You,

Thank you very much for submitting your manuscript "Data-driven learning of structure augments quantitative prediction of biological responses" for consideration at PLOS Computational Biology.

As with all papers reviewed by the journal, your manuscript was reviewed by members of the editorial board and by several independent reviewers. In light of the reviews (below this email), we would like to invite the resubmission of a significantly-revised version that takes into account the reviewers' comments.

The reviewers agree that this is an interesting paper, but that more work is needed to explain the methodology and ground the work in existing approaches.

We cannot make any decision about publication until we have seen the revised manuscript and your response to the reviewers' comments. Your revised manuscript is also likely to be sent to reviewers for further evaluation.

Sincerely,

Samuel V. Scarpino

Academic Editor

PLOS Computational Biology

Stacey Finley

Section Editor

PLOS Computational Biology

I agree with the reviewers that this is an interesting paper, but that more work is needed to explain the methodology and ground the work in existing approaches.

Reviewer's Responses to Questions

**Comments to the Authors:**

Reviewer #1: The review has been uploaded as an attachment.

Reviewer #2: The manuscript by Ha and colleagues describes a new machine learning appraoch, called structure-augmented regression (SAR), that captures phenotypic or fitness landscapes using less data than traditional regression approaches. In the paper they compare their approach to polynomical regression, support vector regression, random forest and k-nearest neighbours. They apply the approach to different data sets of increasing complexity.

Overall, I think that this is an interesting approach that could have wide applicability. I have some suggestions/comments for the improvement of the manuscript.

Major comments:

What is the relationship of this approach to traditional design of experiments (DOE) and Bayesian optimisation. While I understand that the goal may be slightly different, they are both approaches used widely. Bayesian Optimisation also estimates the fitness landscape, for example. Some text should be added in the introduction.

I was confused about the use of direct regression and SVR. While I did eventually figure this out, you should make it clearer to reader as sometimes you use direct regression and sometimes SVR, meaning the same thing.

It wasn’t clear what the distance means here. I think you mean distance from the decision boundary.

Is there an issue with using the data set twice, once for classification and once for regression? I guess if you had enough data you could do these on two samples of the training data points. This could be discussed somewhere.

Minor comments:

Pg6. Figure 1C is referenced after D,E

Pg12 “in comparison to the counterpart”

Reviewer #3: This paper describes a machine learning methods for bioengineering purposes. The innovation is the utilization by the method of the underlying structure of the biological system under investigation. This algorithm is used to infer the phenotypic respose of the system under perturbation. The authors claim that the algorithm can be used to augment a synthetic biology platform and allow for faster exploration of built biological structures and combinatorial experiments for drug design.

This is a very insightful paper, which postulates the existence of a learnable usually lower dimensional) manifold for input-output relationships in biological systems. The manifold itself is built from data by assuming there is lower dimensional boundary differentiating distinct phenotypes, and then by letting the algorithm discover it.

I like this paper, but I think there are a few shortcomings which should be addressed during revision.

Firstly, the definition of "structure" is vague. I interpret is as the existence of a complex, potentially nonlinear mapping between the inputs and outputs of a biological process. The authors do not give a formal definition. as far as I can tell, but rather leave to the readers the intepretation of this fundamental concept. I think the authors should explain more formally, concretely and above all expliclty what the structure of the biological system is, since it is so central to their methodology.

The authors use support vector classification for their boundary discovery, but state that any regression-based classification method would be sufficient. Given the importance of this step, I would like to see more thorough discussion about it, with the limitations embedded in the various choices of algorithm. Right now, this is limited to the appendix, and I would prefer to see it in the main text.

Minor changes:

1. The abstract contains a reference to SVR, which has not been introduced. Is that a typo and the authors meanth SAR instead?

2. Going into the appendix, one is left with more questions than answers by reading the Structure-Assisted Regression section, which is not written having a general application in mind, but rather for an example of cell survival for a drug combination experiment. Is it possible to apply this method in general? What would be the function that someone would need to use to do that?

3. In general, the use of the various acronhyms (SVC, SAR, SVR) is very confusing. Can anything be done about that?

Reviewer #4: In the manuscript titled “Data-driven learning of structure augments quantitative prediction of biological

responses”, Ha et al. proposed a new method of incorporating structural information into regression to predict outcome multi factor induction. The critical contribution is this method would greatly reduce the amount of data needed. I found the results quite interesting and presentation thorough, with some clarification questions:

As I am trying to understand the method,, I found Fig. 1C is a bit confusing (the caption lacks some key details). First, The solid black line is really hard to identify. It took me a while to see, please make that line easier to locate. Second, in Fig. 1C right panel, how the “calculated distance” is calculated? what does “ground truth” mean? The last paragraph on page 6 needs some clarification. All the later figures use prediction and ground truth as axis labels with both scale 0 to 1. How come Fig 1C right panel x-axis range -200 to ~100, with a name calculated distance? Maybe I am missing something here. I would expect a scatter plot of predicted value VS simulated value (ground truth), and compare it to a straight line, thus get a R2 value. Fig 2 C and F bottom right panel show some results as expected. But then in Fig3 the results do no look so good. How come Fig 1G right bottom panel results look like a sigmoidal curve?

**Have the authors made all data and (if applicable) computational code underlying the findings in their manuscript fully available?**

Reviewer #1: **No: **At the moment the GitHub repository, https://github.com/YChH/StructureLearning , is empty.

Reviewer #2: Yes

Reviewer #3: Yes

Reviewer #4: Yes

PLOS authors have the option to publish the peer review history of their article (what does this mean?). If published, this will include your full peer review and any attached files.

Reviewer #1: No

Reviewer #2: No

Reviewer #3: No

Reviewer #4: No
---

## [Decision Letter · Decision Letter 1]

20 May 2024

Dear Dr. You,

We are pleased to inform you that your manuscript 'Data-driven learning of structure augments quantitative prediction of biological responses' has been provisionally accepted for publication in PLOS Computational Biology.

Best regards,

Samuel V. Scarpino

Academic Editor

PLOS Computational Biology

Stacey Finley

Section Editor

PLOS Computational Biology

Reviewer's Responses to Questions

**Comments to the Authors:**

Reviewer #1: I thank the authors for their thorough and thoughtful response to my comments. All my concerns have been addressed in the revised version of the manuscript.

Reviewer #2: The authors have addressed all my comments. The paper is much improved.

Reviewer #3: The authors have successfully addressed all my comments.

Reviewer #4: All my concerns are fully addressed. No more concerns and recommend for publication.

**Have the authors made all data and (if applicable) computational code underlying the findings in their manuscript fully available?**

Reviewer #1: Yes

Reviewer #2: Yes

Reviewer #3: Yes

Reviewer #4: None

PLOS authors have the option to publish the peer review history of their article (what does this mean?). If published, this will include your full peer review and any attached files.

Reviewer #1: No

Reviewer #2: No

Reviewer #3: No

Reviewer #4: No

---

## [Editor Report · Acceptance letter]

30 May 2024

PCOMPBIOL-D-23-01992R1 

Data-driven learning of structure augments quantitative prediction of biological responses

Dear Dr You,

I am pleased to inform you that your manuscript has been formally accepted for publication in PLOS Computational Biology. Your manuscript is now with our production department and you will be notified of the publication date in due course.

With kind regards,

Anita Estes
